RESEARCH ARTICLE *Mem Inst Oswaldo Cruz*, Rio de Janeiro, Vol. *121*: e250154, 2026 1│13

# Behavioural responses of *Anopheles darlingi* (Diptera: Culicidae) to host odours: insights from vertical and horizontal olfactometers

**Thais Costa dos Santos¹, Moreno Magalhães De Souza Rodrigues²,**
**Elis Paula de Almeida Batista³, Kelly da Silva Paixão⁴, Álvaro Eduardo Eiras⁴,**
**Gabriel Zorello Laporta⁵, Alexandre de Almeida e Silva¹/⁺**

¹Universidade Federal de Rondônia, Departamento de Biologia, Porto Velho, RO, Brasil
²Fundação Oswaldo Cruz-Fiocruz, Porto Velho, RO, Brasil
³Ministério da Saúde, Brasília, DF, Brasil
⁴Universidade Federal de Minas Gerais, Instituto de Ciências Biológicas, Departamento de Parasitologia, Belo Horizonte, MG, Brasil
⁵Centro Universitário Faculdade de Medicina do ABC, Santo André, SP, Brasil

**BACKGROUND** *Anopheles darlingi* is the primary vector of malaria in the Americas, particularly in the Amazon, where it thrives in forest margins. This species exhibits considerable flexibility in feeding behaviour, adapting to environmental conditions and host availability. Previous studies on its attraction to human odour have relied mainly on baited traps, with limited research using vertical olfactometry to explore host-seeking behaviour.

**OBJECTIVES** This study aimed to assess the feasibility of using vertical and horizontal olfactometry to investigate the behavioural responses of both wild and laboratory-reared (F1) *An. darlingi* females to human odours. The odours were presented through direct (hands and feet) and indirect (sweat-impregnated synthetic substrates) stimuli.

**METHODS** Wild mosquitoes were collected from Porto Velho and Candeias do Jamari, Brazil, and laboratory-reared (F1) mosquitoes were bred under controlled conditions. A vertical olfactometer was employed to assess short-range attraction, and a dual-choice horizontal olfactometer evaluated host-seeking behaviour. Human odours were obtained from volunteers' hands, feet, and worn socks. Mosquito responses were analysed for attraction, activity, and inactivity, with statistical analysis performed using two-factor analysis of variance (ANOVA).

**FINDINGS** Wild *An. darlingi* mosquitoes showed significantly higher attraction to human odours compared to F1 mosquitoes in both the vertical and horizontal olfactometers. Wild mosquitoes were more attracted to feet and worn socks than F1 mosquitoes, which exhibited low attraction to both stimuli. The preference index (PI) was higher in wild mosquitoes for both hand and sock odours, indicating a stronger attraction to human odours.

**MAIN CONCLUSIONS** Wild *An. darlingi* mosquitoes exhibit a stronger behavioural response to human odours compared to F1 mosquitoes. The use of olfactometry, particularly vertical and horizontal methods, proved effective in studying *An. darlingi* mosquito's host-seeking behaviour and can be applied to further research on vector behaviour and potential control strategies.

Key words: *Anopheles darling* - malaria - mosquito behaviour - olfactometry - host-seeking behaviour

The mosquito *Anopheles darlingi* is the primary vector of malaria in the Americas, with a broad distribution spanning several countries in Central America and most of South America, from Colombia and Venezuela to northern Argentina. This species prefers breeding in natural water bodies such as lakes, streams, and slow-flowing rivers. Given these ecological preferences, *An. darlingi* thrives in the Amazon region, particularly along forest margins, where environmental conditions support its population persistence.[1,2,3] In Brazil, An. darlingi is considered the main vector species involved in the transmission of Plasmodium spp. to humans. This species is highly adaptable to environmental changes such as deforestation and urbanisation.[4,5] Adult *An. darlingi* mosquitoes exhibit considerable plasticity in their feeding behaviour, adapting to local environmental conditions and host availability.[6] They can be found both indoors and outdoors, as well as in peridomestic environments.[7] Adults exhibit crepuscular-nocturnal habits, during which mating occurs and females perform haematophagy. In the laboratory, specific conditions are required to obtain F1 generation mosquitoes from already inseminated wild females collected in the field. The colonisation of this species required overcoming barriers related to the swarming behaviour for mating and was successfully reported in Brazil in 2019.[8] Despite its well-established epidemiological importance as a malaria vector, studies investigating *An. darlingi*'s attraction to human odour have predominantly relied on baited traps. However, the only study to date that assessed this species' behaviour using vertical olfactometry with *Plasmodium*-infected patients.[9]

+ Corresponding author: alealsil@unir.br | ⓘ https://orcid.org/0000-0002-6500-0402

**Handling editor:** Ademir de Jesus Martins Jr | ⓘ https://orcid.org/0000-0001-5739-1215

Olfactometry is a well-established technique for studying the host-seeking behaviour of mosquitoes.[10,11,12,13] It allows researchers to evaluate various behavioural responses, including attraction, activation, and flight pathways in response to chemical (kairomones, *i.e.*, natural or synthetic host odour mimics) and physical (heat, humidity) stimuli.[14,15] Olfactometers have been widely employed to investigate mosquito-host interactions, as well as other behavioural aspects such as feeding,[16,17,18] nutrition,[19,20] and reproduction.[21,22]

Different olfactometry techniques, including vertical[23,24] and horizontal dual-choice[25,26] methods, can assess mosquito responses to potential host stimuli. Vertical olfactometry evaluates mosquito attraction through passive diffusion of olfactory and physical cues (*e.g.*, humidity and heat) without visual interference.[27] This method allows the use of human body parts, such as hands or feet, as odour sources, in addition to synthetic attractants. In contrast, horizontal dual-choice olfactometry exposes mosquitoes to a moving odour plume within a controlled airflow system, enabling anaemotaxic responses. This setup facilitates behavioural categorisation into four responses: (a) active (mosquitoes resting or flying outside the choice tube), (b) inactive (mosquitoes remaining in the release cage or failing to take off), (c) attracted (mosquitoes flying against the airflow inside the choice tube), and (d) take-off (mosquitoes leaving the release cage)[28]

This study aimed to assess the feasibility of using vertical and horizontal olfactometry to investigate the behavioural responses of wild *An. darlingi* females and first-generation (F1) laboratory-reared females to human odours. These odours were presented through direct stimuli (*e.g.*, hands and feet) and indirect stimuli (*e.g.*, sweat-impregnated synthetic substrates).

## SUBJECTS AND METHODS

*Mosquitoes* - Wild females were collected during the crepuscular-nocturnal period (18:00-22:00 h) by protected human landing catches at three sites: two in Porto Velho (Nova Mutum: 9°18′55.51″S, 64°32′44.96″W; BR 364: 8°49′22.81″S, 63°55′42.57″W) and one in Candeias do Jamari (8°47′31.93″S, 63°42′39.79″W). Approximately 20 specimens were maintained in 500 mL plastic screened cages, placed in insulated boxes, and supplied with 10% sucrose solution on soaked cotton pads. Mosquitoes were transported to the insectary of Laboratório de Bioecologia de Insetos da Universidade Federal de Rondônia, Porto Velho, Brazil. Female An. darlingi were identified using a dichotomous key based on morphological.[29] Insectary conditions were 27 ± 1°C, 80 ± 5% relative humidity (RH), and a 12:12 h light-dark cycle. Wild females were tested within 48 h of capture, after 24 h of sugar deprivation with access to water only. A subset of wild females was dissected under a stereomicroscope using fine needles and saline solution to verify the presence of spermatozoa in the spermathecae.

As a comparative group, F1 generation females of An. darlingi without previous blood feeding were employed. These were obtained from wild females captured exclusively for colony establishment. Females were fed on Mus musculus (Swiss inbred strain) and, 72 h after feeding, were induced to oviposit by removal of one wing. Resulting larvae were reared under standard insectary conditions (27 ± 1°C, 80 ± 5% RH, 12:12 h light-dark cycle) and fed ad libitum with fish food (Tetramin Tropical Flakes®, Tetra Holding US Inc., USA). Newly emerged F1 females were housed in entomological cages (29.0 × 20.0 × 7.0 cm) and maintained on 10% sucrose solution. Females aged five-seven days were selected as comparative group (F1), representing unfed and unmated females.

## Olfactometers

*Vertical olfactometer* - This olfactometer[24] is composed of four parts, which are made of aluminium to avoid the retention of odour molecules, being (a) a box open on one side (35.5 cm in length, 34.0 cm in width, 37.0 cm height); (b) a cone placed inside the box (10.0 cm in height and 7.0 cm in diameter); (c) a mesh-covered cone positioned on the upper external part of the box (21.0 cm in height, with a top diameter of 14.4 cm and a bottom diameter of 7.4 cm); and (d) a cylinder placed externally around the mesh-covered cone (26.8 cm in height and 22.0 cm in diameter), which is employed to avoid visual interference during tests (Fig. 1A). This type of olfactometer allows the evaluation of chemical (odours) and/or physical stimuli at a short distance through passive diffusion between the cones of the odour plume formed. To do this, the stimuli to be evaluated is inserted below the lower cone and the insects are released at the top of the screened cone and the response to the stimuli offered is measured by the proportion of insects that move from the upper cone to the lower cone. The vertical distance between mosquitoes in the upper cone and the stimuli ranged from 35.05 to 51.0 cm, depending on insect position. Passage from the upper to the lower cone required overcoming a barrier at the cone junction, ensuring that movement was directed by the stimuli.

*Dual-choice olfactometer* - The dual-choice horizontal olfactometer consists of an experimental arena made of acrylic to evaluate the response of mosquitoes to an odour plume. The olfactometer consists of (a) a removable cylindrical release cage (7.0 cm in diameter and 10.0 cm in length, thickness: 0.5 cm) through which mosquitoes were released; (b) an elongated cylindrical base tube (8.0 cm in diameter and 50.0 cm in length) attached to a central rectangular compartment (28,0 cm in length 16.0 cm in width 14.0cm height); (c) two choice tubes (7.0 cm in diameter, 44.0 cm in length) where the stimuli were inserted (10.0 cm in diameter and 20.0 cm in length); and (d) a choice box connected to the three tubes, connected to the three tubes. The odour plume is generated by an air flow, at a speed of 0.30 m/s, drawn by an exhaust (e) fan from the external environment, which is filtered through a heated/humidified charcoal filter (27 ± 1°C /70% RH) in a water bath and passing the stimuli towards the mosquitoes that are in the release cage (Fig. 1B).

*Selection of volunteers and human odour collection* - Four volunteers (two men and two women, all Caucasian, aged 20-40 years) participated in the study. They were instructed to refrain from using lotions, perfumes, or antiseptic powders on their feet and hands — used as odour sources in the experiments — for four days prior to testing.

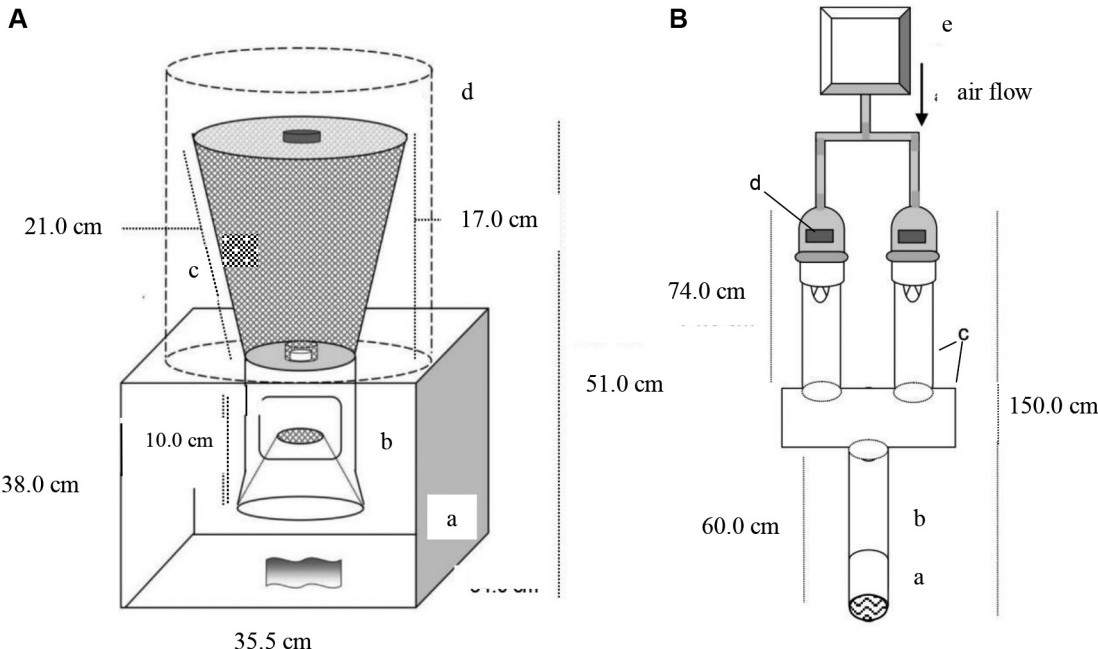

Fig. 1: schematic design of experimental olfactometers (A) Vertical olfactometer: (a) open-sided box for stimuli placement; (b) lower cone containing responding insects; (c) screened upper cone for mosquito release; (d) external cylinder to prevent visual interference. (B) Horizontal dual-choice olfactometer: (a) removable release cage; (b) base tube; (c) two choice tubes with stimuli; (d) opening gap to insert stimuli; (e) exhaust pulls air from the outside environment that is filtered through an activated charcoal filter, heated in a water bath generating a 0.30 m/s airflow [27 ± 1ºC, 70% relative humidity (RH)].

Human odour samples were collected using thin black socks made of 100% polyamide (Selene brand). Before use, the socks were washed with a neutral detergent (Extran, 6% concentration: 60 mL/L), dried with hot air (70ºC) for 10 min, and then worn by each volunteer for 10 h until 30 min before testing. The used socks were stored in zip-lock bags (Wyda Zip brand) until the experiments commenced.

**Olfactometer bioassays**

*Short-distance attractiveness of An. darlingi to human odours using a vertical olfactometer* - To assess short-range attractiveness, experiments were conducted in a vertical olfactometer (modified from Eiras & Jepson)[24] using wild and F1 *An. darlingi* females. Trials began after 6:00 pm, aligning with the mosquitoes' twilight host-seeking behaviour. Each experiment lasted 10 min, with the first 5 min allowing the mosquitoes to acclimate inside the screened cone before the stimuli was introduced beneath the lower cone. The number of mosquitoes that moved to the lower cone was recorded at the end of each trial.

Two experimental conditions were tested:

• Direct source: One foot of each volunteer served as the odour stimuli (treatment), while an empty olfactometer was employed as the control.

• Indirect source: The stimuli (treatment) was a sock worn by each volunteer, with a clean sock as the control.

Four volunteers participated, and five replicates were performed per stimuli (foot or worn sock) with groups of 10 mosquitoes each, for both wild-type and F1 females. A total of 80 trials were conducted, using 800 females. After each experiment, the olfactometer components were cleaned with a 6% Extran detergent solution (Merck brand) and dried with hot air (70ºC).

*Host-seeking behaviour of An. darlingi in response to human odours using a dual-choice horizontal olfactometer* - Wild and F1 *An. darlingi* females were tested in a dual-choice horizontal olfactometer (modified from Paixão et al.)[22] to evaluate host-seeking behaviour in response to human odours. Trials were conducted after 6:00 pm, beginning with a 10 min acclimatisation period in the release cage, followed by 3 min of exposure to the stimuli introduced into one of the choice tubes.

Two experimental conditions were tested, differing in the stimuli employed:

Experiment 1: A volunteer's hand was the stimuli (treatment), while the control was an empty choice tube (negative control).

Experiment 2: A worn sock served as the stimuli (treatment), with the control being an empty choice tube (negative control).

Two additional volunteers participated in these experiments. Seven replicates were performed per volunteer per stimuli (hand or worn sock) with groups of 10 mosquitoes each (wild or F1 females). In total, 56 trials were conducted, using 560 mosquitoes.

Following Paixão et al.,[22] mosquito responses were categorised as:

(a) Attracted: Mosquitoes that entered the choice tubes.

(b) Active: Mosquitoes that left the release cage but did not enter the choice tubes.

(c) Inactive: Mosquitoes that remained in the release cage.

After each experiment, olfactometer components

were washed with a 6% Extran detergent solution, air-dried, then cleaned with 70% ethanol and dried again with hot air (70ºC).

*Statistical snalysis* - In vertical olfactometer experiments, the percentage of mosquitoes that responded (*i.e.*, collected in the lower cone) to the stimuli — either the odour of volunteers' feet or worn socks — was analysed using a two-factor analysis of variance (ANOVA) (mosquito strain × stimuli), followed by Sidak's post hoc test at a 5% significance level. The control condition was an olfactometer without stimuli.

In dual-choice horizontal olfactometer experiments, the percentage of mosquitoes in each response category — Attracted (entered the treatment arm), Control (entered the control arm), Inactive (remained in the release cage), and Active (left the release cage but did not enter either arm) — was analysed using a two-factor ANOVA (mosquito strain × behaviour), with post hoc comparisons using the Sidak test at a 5% significance level.

The preference index (PI) was calculated using the formula:

$$ PI = \frac{(\text{Number of mosquitoes in the treatment arm - Number of mosquitoes in the control arm})}{(\text{Number of mosquitoes in the treatment arm + Mosquito number in control arm})} $$

The PI ranges from 0 to 1, where 0 indicates no preference (neutrality) and 1 indicates maximum attraction. Data were analysed using a two-factor ANOVA (mosquito strain × stimuli), with post hoc comparisons using the Sidak test at a 5% significance level.

*Ethical considerations* - All volunteers were fully informed about the study's objectives, potential benefits, and risks before providing written informed consent. Ethical approval was granted by the Human Research Ethics Committee of the Research Centre for Tropical Medicine of Rondônia (CAAE No. 41977315200000011).

The use of mice for blood-feeding followed ethical guidelines approved by CEUA/FIOCRUZ-RO (Protocol No. 2014/15).

## RESULTS

*Assessment of An. darlingi attractiveness to human odours in a vertical olfactometer* - Overall, both feet and used socks attracted more mosquitoes than the empty olfactometer (control) (F = 9.11, p = 0.0004). Wild *An. darlingi* mosquitoes showed a significantly higher attraction to both feet (23%) and used socks (17%) compared to F1 mosquitoes (8% and 4%, respectively) (Feet: F = 7.8, p = 0.0078; Socks: F = 10.9, p = 0.0021). The volunteers' feet attracted approximately 11 times more wild mosquitoes (23%) than the control condition (empty olfactometer, 2%). In contrast, for F1 mosquitoes, no significant differences were observed between the attraction to feet (8%) and the control (1%), nor between the attraction to used socks (4%) and the control (2%) (p > 0.05) (Fig. 2).

When comparing a used sock versus a clean sock (control) in the same olfactometer, no significant difference was found in the proportion of F1 mosquitoes attracted (p > 0.05). However, for wild mosquitoes, the used sock attracted a significantly higher proportion (16%) compared to the clean sock (5%) (p < 0.05).

*Evaluation of host-seeking behaviour of An. darlingi exposed to human odours using a dual-choice horizontal olfactometer* - F1 mosquitoes showed a significantly higher percentage of inactivity (43.3%) compared to activation (27.8%) when the hand was used as a stimuli (F = 26.3, p < 0.0001). However, no significant differences were observed between activation and inactivity in the other experimental groups and treatments (p > 0.05) (Fig. 3A).

Regardless of the stimuli source or mosquito strain, the percentage of mosquitoes attracted to the stimuli arm was consistently higher than to the control arm.

When hands were employed as stimuli, the attraction rates were 28.8% (F1) and 39.9% (wild-type), while the control arms had significantly lower attraction rates (8.3% for F1, p = 0.0008; 6.5% for wild-type, p < 0.0001).

When used socks were the stimuli, attraction rates were 27.4% (F1) and 42.3% (wild-type), whereas the control arms had 10.3% (F1, p = 0.0016) and 11.6% (wild-type, p < 0.0001) (Fig. 3B).

Inactive: mosquitoes that remained in the release cage. Active: mosquitoes that left the release cage but did not enter the olfactometer arms. Control: mosquitoes attracted to the control arm (without stimuli). Attracted: mosquitoes attracted to the arm with the stimuli.

The PI of wild *An. darlingi* in dual-choice horizontal olfactometry experiments was significantly higher (F = 6.7; p = 0.01) combining both stimuli — hand and used socks (0.63) — compared to F1 mosquitoes (0.33). However, no significant difference was observed between the responses to hand and sock odours within each mosquito group (F = 3.4; p = 0.07) (Fig. 4).

After examining the spermathecae of a pool of wild An. darlingi females from the collection sites, we found that 93/97 contained sperm, two were empty, and two were lost during dissection.

## DISCUSSION

Olfactometers can be employed to evaluate the oviposition preferences and attraction of *Anopheles* mosquitoes to breeding sites. They also offer the advantage of detecting significant substrates for creating control

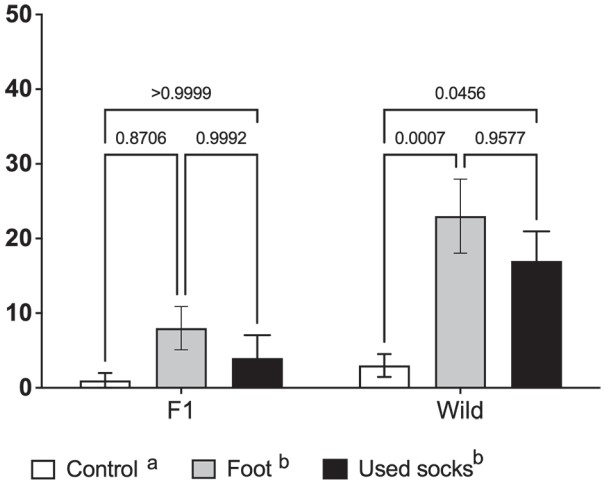

Fig. 2: close-range attractiveness of wild and F1 *Anopheles darlingi* to human odours, *i.e.*, feet and used socks, using a vertical olfactometer.

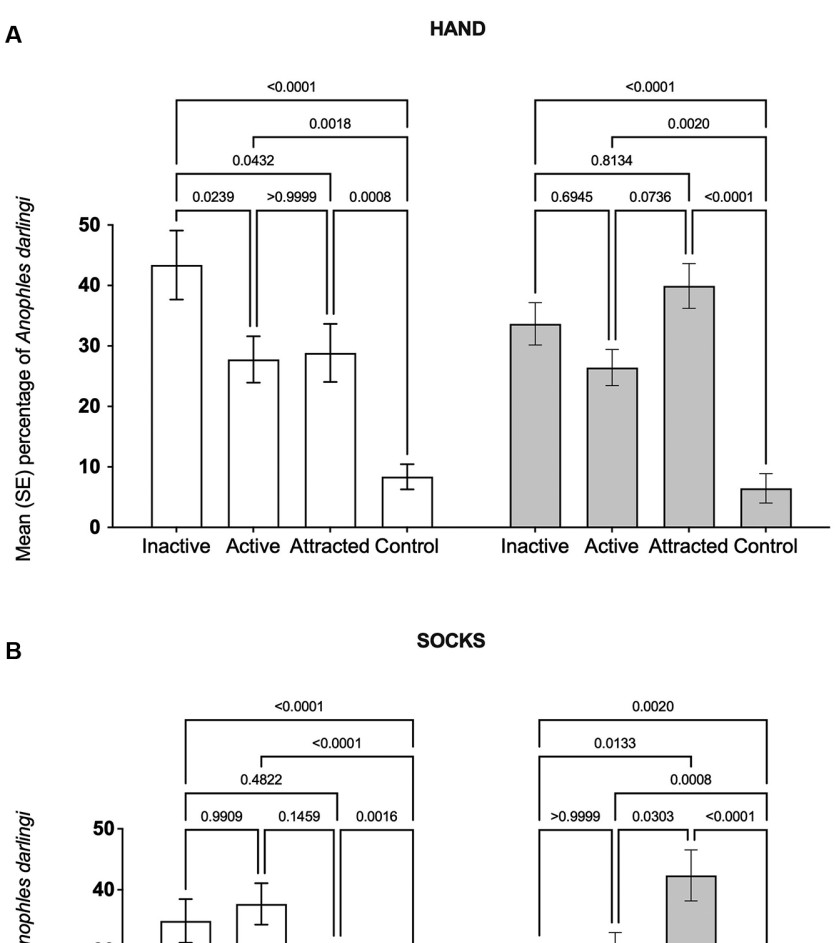

Fig. 3: behaviour of wild *Anopheles darlingi* and F1 in a dual-choice horizontal olfactometry assay, using hands and used socks as olfactory stimuli.

measures that can modulate the behaviour of gravid females. However, the use of these instruments has some limitations, such as the restricted number of individuals that can be analysed simultaneously and the need for objective quantification methods.[30]

The use of vertical olfactometry with used socks, as well as volunteers' feet as stimuli in this study, was effective in attracting *An. darlingi* females, but only when wild mosquitoes were employed. The use of sweat-impregnated socks has been successfully applied to attract other mosquito genera, such as *Aedes* and *Culex*.[31,32] However, this is the first successful use of these stimuli for this important Neotropical vector.

In terms of host odour sources, used socks attracted *An. darlingi* females similarly to volunteers' feet, with no significant differences between these two sources.

The results suggest that the used socks effectively captured volatile attractants or components thereof. However, only field-collected females were significantly attracted to the impregnated sock odour sources. The use of socks as an odour source in experiments with *An. darlingi* was effective and supports findings by Okumu et al.[33] on the effectiveness of synthetic materials (*e.g.*, nylon) for dispersing kairomones in traps for other anopheline species. Additionally, using socks impregnated with odours from patients with symptomatic diseases could enhance research, as patient compliance with travel to laboratories for experiments is often low.[9,16,33] In this study, both wild *An. darlingi* females and F1 females were significantly attracted to the odours of the host on impregnated socks and volunteers' hands (attracted) compared to the control (empty arm). The ma-

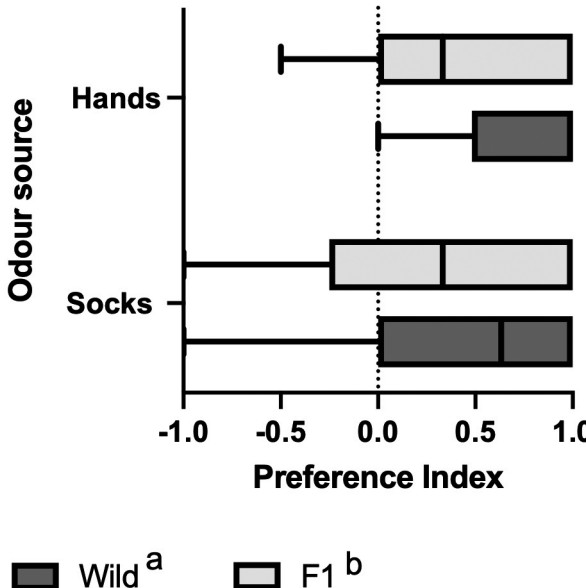

Fig. 4: preference index (PI) of wild and F1 *Anopheles darlingi* for hand and used sock odour stimuli from volunteers during dual-choice horizontal olfactometry experiments. Different letters indicate significant differences between treatments (p < 0.05).

jority of wild *An. darlingi* females in the present study were inseminated as suggested by our data from a pool of wild females collected in the field. Both inseminated and virgin An. darlingi females are highly active during the scotophase, likely reflecting blood-seeking in the former and mating swarm formation in the latter.[34] Thus, although both wild and F1 females were expected to be active during the assays, mostly wild (inseminated) females would respond to human odour, consistent with their physiological need for a blood meal to support egg maturation. Similarly, Paixão et al.[22] reported that both virgin and mated *Ae. aegypti* females were significantly attracted to human hand odours compared to controls (empty cages), although virgin females were generally less attracted than mated females.

Glass spheres impregnated with hand odour were able to attract *An. quadrimaculatus* and *Ae. aegypti* females in a dose-dependent manner in a dual-choice horizontal olfactometer, but were not directly compared to hand odour.[35] In studies conducted in Kenya, traps baited with used socks collected in huts did not show a difference in capturing *An. gambiae* or *An. funestus* compared to human collection.[11] Similar results were observed in a study on *An. gambiae* conducted in western Kenya, where no significant difference was found between mosquitoes captured in traps baited with impregnated socks and human odour.[36] However, individual differences in volatile profiles, including between sexes, should not be disregarded.[37]

Host-seeking behaviour in female Anopheles mosquitoes is strongly modulated by their physiological state. Age, body size, nutritional reserves, and gonotrophic status all influence the likelihood and intensity of host-seeking responses.[38,39,40] Importantly, some Anopheles species exhibit opportunistic host seeking even before mating, indicating that reproductive and nutritional demands may overlap during the early adult phase.[41] Thus, the responsiveness of females to semiochemicals such as $CO_2$ and human-derived volatiles is not fixed, but varies dynamically according to their physiological condition.[42,43]

In our comparison of the two olfactometer designs, it is important to note that the vertical olfactometer relies on passive diffusion of host odour, which results in slower accumulation of volatiles and delayed mosquito responses, often taking several minutes after the stimuli is introduced. In contrast, dual-port horizontal olfactometers employ a forced and filtered airflow that continuously delivers host odours in a controlled manner, producing a more uniform odour plume and eliciting rapid, upwind flight responses.[15] Therefore, it is expected that response rates in the horizontal olfactometer will generally exceed those obtained in the vertical design, as the former better reproduces the ecological conditions under which mosquitoes locate their hosts.

Our findings demonstrate that wild An. darlingi females exhibited significantly stronger behavioural responses to human odours than laboratory-reared F1 virgins, highlighting the influence of physiological state on host-seeking activity. Both vertical and horizontal olfactometers proved to be effective tools for assessing mosquito attraction to direct (hands, feet) and indirect (worn socks) human odour stimuli, except in the vertical olfactometer using F1 mosquitoes. Worn socks performed comparably to human feet, supporting their use as standardised odour sources in laboratory assays. Overall, olfactometry offers a valuable approach for advancing behavioural studies of An. darlingi and provides insights that may contribute to the development of odour-based monitoring and control strategies for this malaria vector.

## ACKNOWLEDGEMENTS

To the Entomology Department at Fiocruz Rondônia for their support in the logistics of anopheline collections and for providing insectaries to maintain mosquitoes.

## AUTHORS' CONTRIBUTION

AAS - project administration, supervision, wrote - original draft, writing review & editing; AEE - conceptualisation, resources, writing - review & editing; KSP - methodology, supervision and writing - review & editing; EPAB - methodology and writing - review & editing; MMSR - formal analysis, writing - review & editing; TCS - investigation, methodology, wrote - original draft, writing - review & editing; GZL - writing - review & editing. The authors declare no conflict of interest.

## DATA AVAILABILITY

The contents underlying the research text are included in the manuscript.

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

# OPEN PEER REVIEW

Memórias do IOC thanks the anonymous reviewers for their contribution to the peer review of this work.

## FIRST REVIEW ROUND

<div align="right">REVIEWERS' COMMENTS</div>

### REVIEWER #1

The protocol by dos Santos et al. describes the use of vertical and horizontal olfactometers to study attraction to host odors in Anopheles darlingi, an important vector of malaria in the Americas. The protocol compares attraction of wild and laboratory-reared (F1) mosquitoes to feet, hands, or sweat-impregnated socks using vertical (single-choice) and horizontal (dual-choice) olfactometers. In their manuscript, the authors demonstrate that both olfactometer designs can be used to quantify attraction of this vector species to human odor, particularly in wild-caught mosquitoes. This represents a useful step toward establishing standardized behavioral assays for this important vector species. They also report significant differences between wild and lab-reared mosquitoes, with F1 individuals displaying no or substantially reduced attraction in these assays. However, key aspects of the methodology limit the interpretability of these comparisons. In particular, the laboratory (F1) mosquitoes are first-generation progeny of wild-caught females, and it is unclear whether their reduced attractiveness to human odor comes from a substantially reduced fitness. Their limited activity and poor performance raise concerns about their behavioral competence since no fitness assays were performed and may question the validity of comparing them directly with wild mosquitoes obtained in this study. These issues complicate the interpretation of the data and weaken the conclusions drawn from the comparison between strains. In addition, Figure 2A is missing in the current version of the manuscript, leaving the analysis incomplete and further limiting assessment of the study's conclusions.

Major Comments:

1. The authors report significant differences in odor attraction between the two mosquito groups across both olfactometer types. One striking pattern in the data is the low activity and host-seeking behavior in the F1 mosquitoes. This suggests that these first-generation lab-reared mosquitoes may suffer from reduced fitness, potentially as a result of stress from colonization, mating limitations, or rearing conditions. Their poor responsiveness calls into question whether they are valid representatives of An. darlingi behavior. A basic analysis of locomotor activity could help determine whether the F1 mosquitoes are capable of normal flight and orientation. Alternatively, establishing a more stable, multi-generational laboratory colony might yield more behaviorally robust individuals for comparison. As it stands, the comparison between wild mosquitoes and these F1 individuals—who may be behaviorally compromised—limits the strength and relevance of the study's conclusions.

2. Another major point concerns the mating status of the mosquitoes. While the authors address this in the discussion, its implications for the results are severely underestimated. The authors state that "F1 females do not copulate spontaneously in the laboratory and require external stimuli like light or temperature changes." In addition to the potentially lower fitness of the F1 mosquitoes, differences in reproductive state between wild and F1 females could explain the behavioral results and render direct comparisons between the two groups invalid. Dissection of F1 spermathecae (as done for wild mosquitoes) and oviposition assays could provide insight into how reproductively active F1 females were and thus clarify the impact of reproductive state on host-seeking behavior.

3. Wild mosquitoes were collected using human landing catches. This method biases capture toward females that are actively host-seeking, as clearly reflected in the olfactometer assays. However, this collection bias likely increases the behavioral differences observed between wild and F1 mosquitoes, which may be in very different physiological states.

4. While the authors provide detailed information on the rearing conditions for lab-reared mosquitoes, there is little description of the handling and maintenance of wild mosquitoes. How long after collection were they tested? Were females housed with males to increase the likelihood of mating?

5. A key piece of information missing from the manuscript is the dimensions of the olfactometers. The authors refer to the vertical olfactometer as a short-range host-seeking assay, but without dimensions, it is difficult to determine how short-range the assay truly is.

6. The legend for Figure 1 is incomplete. It refers to a part A and part B that are not labeled on the figure. While the panels are visually labeled, the legend does not explain what each part represents. This is especially problematic for the right panel. It appears that purified environmental air is divided among three olfactometers, but this is not described in the text or legend and should not be left to the reader's interpretation.

7. Why was the control for the horizontal olfactometer an empty tube, rather than an unworn sock as in the vertical olfactometer? Based on the data presented, the horizontal olfactometer appears more effective in attracting F1 mosquitoes. However, the use of an empty tube as a control (instead of an unworn sock) may have increased the preference index observed in these experiments.

8. Panel A of Figure 2 appears to be missing. The plot shown lacks a label on the y-axis, and the meaning of the error bars is not stated.

9. In Figure 4, several statistical comparisons mentioned in the text are not reflected in the plot. Direct comparisons (e.g., Hands: wild vs. F1; Socks: wild vs. F1) should be clearly indicated with appropriate statistical annotations.

Minor Comments:

1. In line 22 of the abstract, the authors state "particularly in the context of Plasmodium-infected hosts." It is unclear why this is mentioned, and it misleads the reader into expecting that host-seeking in infected hosts will be examined.

2. Line 283: Of the mosquitoes tested, what proportion of females were positive for sperm in the spermathecae? This would provide valuable information on the reproductive state of the wild mosquitoes tested in the assays.

3. Is there a reason why the acclimation period was 5 minutes in the vertical olfactometer but only 1 minute in the horizontal olfactometer?

4. Line 269: The authors state, "Overall, An. darlingi females were less attracted to olfactory stimuli compared to Aedes aegypti." This comparison is not valid and should be removed, as it is based on data from different species and different studies, under potentially different experimental conditions.

**REVIEWER #2**

The abstract is adequate and reflects the findings of the study. Although horizontal olfactometer assays have been existing for a while, the novelty of the paper is the push for vertical olfactometer as a compliment or alternative way to test host seeking. The Methodology and data analysis were thorough and leave no bias to doubt the findings or the reproducibility of the experiments.

Minor comments: all generic names should be italicized. For example, line 58 - plasmodium.

**REVIEWER #3**

The paper is very interesting and the methodology is sound. I have a few comments to improve the manuscript.

Please add a paragraph to the introduction explaining a bit more why Anopheles darlingi is difficult to work with. Some of this is in the discussion, but a brief summary in the introduction would give the reader more context. For example, the mating issues in the lab need to clearly stated in the introduction.

I would also add a sentence or two more on the importance of the species. I feel it is there in the introduction, but casual readers may miss it.

The image quality and size of the olfactometer figure could be improved. Make sure the resolution and text are large enough to be clear in the published manuscript.

I feel this is important and difficult work that will help build the foundation to study this species behavior.

## AUTHORS' RESPONSE TO THE REVIEWERS

Dear Dr Martins Jr.

I'm attaching the revised version of our article entitled "Behavioral responses of Anopheles darlingi (Diptera: Culicidae) to host odors: insights from vertical and horizontal olfactometers)" We have now completed the revision in the manuscript and responded (italics) to each of the referee's questions or comments:

Editor: However, before the manuscript can be considered for publication, several substantial issues need to be addressed. These include concerns regarding the interpretation of behavioral differences between wild and F1 mosquitoes—particularly in light of potential differences in fitness and reproductive status—as well as the absence of key methodological details and incomplete figure presentation. We invite you to revise your manuscript thoroughly, addressing the methodological concerns, clarifying experimental conditions, and improving figure quality and consistency. Please also consider reinforcing the introduction with additional context on the biological relevance and experimental challenges associated with An. darlingi.

Authors: We agree with the editor's comments and have restructured the text by incorporating new discussions. We added information in the introduction about the biological relevance of the species and the reproductive challenges of working with it. Regarding the behavioral differences between wild and F1 females, we have addressed them accordingly.

Referee 1: The authors report significant differences in odor attraction between the two mosquito groups across both olfactometer types. One striking pattern in the data is the low activity and host-seeking behavior in the F1 mosquitoes. This suggests that these first-generation lab-reared mosquitoes may suffer from reduced fitness, potentially as a result of stress from colonization, mating limitations, or rearing conditions. Their poor responsiveness calls into question whether they are valid representatives of An. darlingi behavior. A basic analysis of locomotor

activity could help determine whether the F1 mosquitoes are capable of normal flight and orientation. Alternatively, establishing a more stable, multi-generational laboratory colony might yield more behaviorally robust individuals for comparison. As it stands, the comparison between wild mosquitoes and these F1 individuals—who may be behaviorally compromised—limits the strength and relevance of the study's conclusions.

Authors: We understand the referee's concern regarding the factors that may have led to the differences we observed between wild and F1 females. The difference in host-seeking behavior between the two lineages was expected, since the former are found mostly inseminated, as suggested by the literature and by dissections of a subsample of wild females in our study, while the latter are virgins. Considering our rearing conditions, similar to those used even in colonies that are now established, we believe the differences did not result from reduced fitness but rather from the physiological status of the lineages. We have included a paragraph in the discussion to address the impact of physiological state on host-seeking behavior. A locomotor activity study conducted with colonized An. darlingi showed that both inseminated and virgin females were highly active during the scotophase (Bastos et al., 2023). We discussed the differences between wild and F1 females using this information. In conclusion, with the changes made and considering the limitations of the study, we believe our most important conclusion—that olfactometers can be used for the study of An. darlingi—remains valid, and that, by considering and discussing the differences between field and F1 females, these results provide insights that can be applied in future assays with now-colonized mosquitoes.

Referee 1: Another major point concerns the mating status of the mosquitoes. While the authors address this in the discussion, its implications for the results are severely underestimated. The authors state that "F1 females do not copulate spontaneously in the laboratory and require external stimuli like light or temperature changes." In addition to the potentially lower fitness of the F1 mosquitoes, differences in reproductive state between wild and F1 females could explain the behavioral results and render direct comparisons between the two groups invalid. Dissection of F1 spermathecae (as done for wild mosquitoes) and oviposition assays could provide insight into how reproductively active F1 females were and thus clarify the impact of reproductive state on host-seeking behavior.

Authors: To provide context for the time when this work was conceived (2015), there were no established An. darlingi colonies available in Brazil (Araujo et al., 2019). Therefore, to obtain mosquitoes for medical entomology research, we routinely generated the F1 generation from wild-caught females collected in rural areas around Porto Velho, which were maintained in the insectary of the Fiocruz Rondônia Entomology Laboratory. Regarding the examination of spermathecae of virgin females, we considered it unnecessary since these were reared insects and not from a colony. This assumption was supported by the inability of An. darlingi to copulate in small cages (eurigamic behavior) unless colonies are established, which was not the case in our study.

Referee 1: Wild mosquitoes were collected using human landing catches. This method biases capture toward females that are actively host-seeking, as clearly reflected in the olfactometer assays. However, this collection bias likely increases the behavioral differences observed between wild and F1 mosquitoes, which may be in very different physiological states.

Authors: Unfortunately, at the time this study was conducted, An. darlingi colonies were not yet available in Brazil, and the standard collection method for this species in the field—still in use today—is protected human landing catches. Indeed, as already mentioned and discussed in the paper, over 95% of field-collected females were inseminated, whereas F1 females in the laboratory were virgins, given that copulation does not occur under the rearing conditions used. Therefore, the differences between wild and F1 females likely resulted from their distinct physiological states and their specific responses to host odors, as we verified in our assays. Nevertheless, a recent study by Bastos et al. (2023), conducted with colonized An. darlingi, showed that both inseminated and virgin females were highly active during the scotophase. We discussed the differences between wild and F1 females using this information.

Referee 1: While the authors provide detailed information on the rearing conditions for lab-reared mosquitoes, there is little description of the handling and maintenance of wild mosquitoes. How long after collection were they tested? Were females housed with males to increase the likelihood of mating?

Authors: They were tested approximately 48 hours after collection. Only females were collected in the field, since protected human landing catches do not allow male capture. As we reported in one sample, about 95% of field-collected females were already inseminated. Information about the handling and maintenance of wild mosquitoes has now been added to the Methods section.

Referee 1: A key piece of information missing from the manuscript is the dimensions of the olfactometers. The authors refer to the vertical olfactometer as a short-range host-seeking assay, but without dimensions, it is difficult to determine how short-range the assay truly is.

Authors: We apologize for the lack of detail. The dimensions have been included in the figures, based on the original articles used to construct the olfactometers.

Referee 1: The legend for Figure 1 is incomplete. It refers to a part A and part B that are not labeled on the figure. While the panels are visually labeled, the legend does not explain what each part represents. This is especially problematic for the right panel. It appears that purified environmental air is divided among three olfactometers, but this is not described in the text or legend and should not be left to the reader's interpretation.

Authors: The legend for Figure 1 has been revised, and we included the description of each part as requested. We removed the panel showing the three olfactometers to focus on the part that better explains the horizontal olfactometer itself.

Referee 1: Why was the control for the horizontal olfactometer an empty tube, rather than an unworn sock as in the vertical olfactometer? Based on the data presented, the horizontal olfactometer appears more effective in attracting F1 mosquitoes. However, the use of an empty tube as a control (instead of an unworn sock) may have increased the preference index observed in these experiments.

Authors: We followed the methodology proposed by Paixão et al. (2015), who used an empty tube as the negative control. In the vertical olfactometer assays, although the negative control is also an empty arena, we used a clean sock to ensure there was no interference from its components, but we observed no difference compared to the empty arena. Regarding the higher attraction of F1 mosquitoes in the horizontal olfactometer, we believe the empty tube was not responsible for the variation, since greater activation is expected in this device due to the airflow generating an odor plume, while in the vertical olfactometer odor diffuses passively to the upper tube. The horizontal olfactometer better simulates the natural behavior of mosquitoes, which involves oriented flight against the airflow. Despite these "differences," the overall pattern—i.e., higher responses of wild mosquitoes compared to F1—was maintained. We added a paragraph at the end of the discussion to better address this issue.

Referee 1: Panel A of Figure 2 appears to be missing. The plot shown lacks a label on the y-axis, and the meaning of the error bars is not stated.

Authors: We apologize. We submitted the wrong figure during submission. The correct figure has been added with the appropriate y-axis label.

Referee 1: In Figure 4, several statistical comparisons mentioned in the text are not reflected in the plot. Direct comparisons (e.g., Hands: wild vs. F1; Socks: wild vs. F1) should be clearly indicated with appropriate statistical annotations.

Authors: We found differences only in the overall comparison between wild and F1 mosquitoes, which was indicated by different letters next to the figure legend. We did not observe specific differences between the pairs Hands: wild vs. F1; Socks: wild vs. F1, and therefore no statistical notation was added for them. We revised the text to better reflect this result.

Referee 1: In line 22 of the abstract, the authors state "particularly in the context of Plasmodium-infected hosts." It is unclear why this is mentioned, and it misleads the reader into expecting that host-seeking in infected hosts will be examined.

Authors: We removed this part of the sentence to avoid confusion for the reader.

Referee 1: Line 283: Of the mosquitoes tested, what proportion of females were positive for sperm in the spermathecae? This would provide valuable information on the reproductive state of the wild mosquitoes tested in the assays.

Authors: We did not analyze the spermatheca content of all wild females used in olfactometer assays. However, we were able to carry out this analysis in a subsample of wild females collected from the same areas and observed spermatozoa in 93/97 spermathecae. This information has been inserted in the last paragraph of the Results section, along with contextualization in the Discussion.

Referee 1: Is there a reason why the acclimation period was 5 minutes in the vertical olfactometer but only 1 minute in the horizontal olfactometer?

Authors: Few studies have been published testing the attractiveness of culicids in vertical olfactometers using human body parts. We based our method on the study published by Feinsod and Spielman (1979) on the attraction of Aedes aegypti to the human hand. That study used 5 minutes as a pre-test acclimatization period, and we adapted the exposure time from 7 to 10 minutes because personal observations made in the insectary suggested that An. darlingi is a more timid mosquito compared to Ae. aegypti. In the manuscript, there was a typographical error regarding the acclimatization period for the dual-choice olfactometer: where it read "one-minute acclimatization period," it should have stated "ten-minute acclimatization period." We followed the methodology of Paixão et al. (2015), with adaptations for An. darlingi. We corrected the acclimatization period in the Materials and Methods section.

Referee 1: Line 269: The authors state, "Overall, An. darlingi females were less attracted to olfactory stimuli compared to Aedes aegypti." This comparison is not valid and should be removed, as it is based on data from different species and different studies, under potentially different experimental conditions.

Authors: We removed this comparison.

Referee 2: All generic names should be italicized. For example, line 58 - Plasmodium.

Authors: We carefully reviewed and corrected all scientific terms.

Referee 3: Please add a paragraph to the introduction explaining a bit more why Anopheles darlingi is difficult to work with. Some of this is in the discussion, but a brief summary in the introduction would give the reader more context. For example, the mating issues in the lab need to clearly stated in the introduction.

Authors: We added information about the biology of An. darlingi in the Introduction.

Referee 3: I would also add a sentence or two more on the importance of the species. I feel it is there in the introduction, but casual readers may miss it.

Authors: We agree with Referee 3 and added information in the introduction on the importance of the species in Brazil and its activity patterns.

Referee 3: The image quality and size of the olfactometer figure could be improved. Make sure the resolution and text are large enough to be clear in the published manuscript.

Authors: We improved the quality, size, and information of the olfactometer figures.

Thank you for your time, help and interest. Many thanks to referee 1 for his/her detailed revision and suggestion to improve our manuscript.

This is an important publication for us; therefore, I look forward to hearing from you.

Sincerely,
Alexandre Silva

## SECOND REVIEW ROUND

**REVIEWERS' COMMENTS**

### REVIEWER #1

The authors addressed all my concerns and I recommend this manuscript for publication. Please ensure that Anopheles darlingi is italicized in all figure legends.

### REVIEWER #2

I feel the authors have addressed my questions adequately. I have two further suggestions to improve the manuscript.

1. The authors should add the sentence starting on line 51 to the previous paragraph. It is better not to have a paragraph fragment.

2. In figure 4, the statistical significance (letters) should be put next to the data not in the legend. The letters should also be capitalized.

Otherwise, the manuscript is ready for publication.

### REVIEWER #3

No comments.

