## [Reviewer Report · FIRST REVIEW ROUND - REVIEWERS COMMENTS]

## REVIEWER #1

The protocol by dos Santos et al. describes the use of vertical and horizontal olfactometers to study attraction to host odors in Anopheles darlingi, an important vector of malaria in the Americas. The protocol compares attraction of wild and laboratory-reared (F1) mosquitoes to feet, hands, or sweat-impregnated socks using vertical (single-choice) and horizontal (dual-choice) olfactometers. In their manuscript, the authors demonstrate that both olfactometer designs can be used to quantify attraction of this vector species to human odor, particularly in wild-caught mosquitoes. This represents a useful step toward establishing standardized behavioral assays for this important vector species. They also report significant differences between wild and lab-reared mosquitoes, with F1 individuals displaying no or substantially reduced attraction in these assays. However, key aspects of the methodology limit the interpretability of these comparisons. In particular, the laboratory (F1) mosquitoes are first-generation progeny of wild-caught females, and it is unclear whether their reduced attractiveness to human odor comes from a substantially reduced fitness. Their limited activity and poor performance raise concerns about their behavioral competence since no fitness assays were performed and may question the validity of comparing them directly with wild mosquitoes obtained in this study. These issues complicate the interpretation of the data and weaken the conclusions drawn from the comparison between strains. In addition, Figure 2A is missing in the current version of the manuscript, leaving the analysis incomplete and further limiting assessment of the study’s conclusions.

Major Comments:

1. The authors report significant differences in odor attraction between the two mosquito groups across both olfactometer types. One striking pattern in the data is the low activity and host-seeking behavior in the F1 mosquitoes. This suggests that these first-generation lab-reared mosquitoes may suffer from reduced fitness, potentially as a result of stress from colonization, mating limitations, or rearing conditions. Their poor responsiveness calls into question whether they are valid representatives of An. darlingi behavior. A basic analysis of locomotor activity could help determine whether the F1 mosquitoes are capable of normal flight and orientation. Alternatively, establishing a more stable, multi-generational laboratory colony might yield more behaviorally robust individuals for comparison. As it stands, the comparison between wild mosquitoes and these F1 individuals—who may be behaviorally compromised—limits the strength and relevance of the study’s conclusions.

2. Another major point concerns the mating status of the mosquitoes. While the authors address this in the discussion, its implications for the results are severely underestimated. The authors state that “F1 females do not copulate spontaneously in the laboratory and require external stimuli like light or temperature changes.” In addition to the potentially lower fitness of the F1 mosquitoes, differences in reproductive state between wild and F1 females could explain the behavioral results and render direct comparisons between the two groups invalid. Dissection of F1 spermathecae (as done for wild mosquitoes) and oviposition assays could provide insight into how reproductively active F1 females were and thus clarify the impact of reproductive state on host-seeking behavior.

3. Wild mosquitoes were collected using human landing catches . This method biases capture toward females that are actively host-seeking, as clearly reflected in the olfactometer assays. However, this collection bias likely increases the behavioral differences observed between wild and F1 mosquitoes, which may be in very different physiological states.

4. While the authors provide detailed information on the rearing conditions for lab-reared mosquitoes, there is little description of the handling and maintenance of wild mosquitoes. How long after collection were they tested? Were females housed with males to increase the likelihood of mating?

5. A key piece of information missing from the manuscript is the dimensions of the olfactometers. The authors refer to the vertical olfactometer as a short-range host-seeking assay, but without dimensions, it is difficult to determine how short-range the assay truly is.

6. The legend for Figure 1 is incomplete. It refers to a part A and part B that are not labeled on the figure. While the panels are visually labeled, the legend does not explain what each part represents. This is especially problematic for the right panel. It appears that purified environmental air is divided among three olfactometers, but this is not described in the text or legend and should not be left to the reader’s interpretation.

7. Why was the control for the horizontal olfactometer an empty tube, rather than an unworn sock as in the vertical olfactometer? Based on the data presented, the horizontal olfactometer appears more effective in attracting F1 mosquitoes. However, the use of an empty tube as a control (instead of an unworn sock) may have increased the preference index observed in these experiments.

8. Panel A of Figure 2 appears to be missing. The plot shown lacks a label on the y-axis, and the meaning of the error bars is not stated.

9. In Figure 4, several statistical comparisons mentioned in the text are not reflected in the plot. Direct comparisons (e.g., Hands: wild vs. F1; Socks: wild vs. F1) should be clearly indicated with appropriate statistical annotations.

Minor Comments:

1. In line 22 of the abstract, the authors state “particularly in the context of Plasmodium-infected hosts.” It is unclear why this is mentioned, and it misleads the reader into expecting that host-seeking in infected hosts will be examined.

2. Line 283: Of the mosquitoes tested, what proportion of females were positive for sperm in the spermathecae? This would provide valuable information on the reproductive state of the wild mosquitoes tested in the assays.

3. Is there a reason why the acclimation period was 5 minutes in the vertical olfactometer but only 1 minute in the horizontal olfactometer?

4. Line 269: The authors state, “Overall, An. darlingi females were less attracted to olfactory stimuli compared to Aedes aegypti.” This comparison is not valid and should be removed, as it is based on data from different species and different studies, under potentially different experimental conditions.

## REVIEWER #2

The abstract is adequate and reflects the findings of the study. Although horizontal olfactometer assays have been existing for a while, the novelty of the paper is the push for vertical olfactometer as a compliment or alternative way to test host seeking. The Methodology and data analysis were thorough and leave no bias to doubt the findings or the reproducibility of the experiments.

Minor comments: all generic names should be italicized. For example, line 58 - plasmodium.

## REVIEWER #3

The paper is very interesting and the methodology is sound. I have a few comments to improve the manuscript.

Please add a paragraph to the introduction explaining a bit more why Anopheles darlingi is difficult to work with. Some of this is in the discussion, but a brief summary in the introduction would give the reader more context. For example, the mating issues in the lab need to clearly stated in the introduction.

I would also add a sentence or two more on the importance of the species. I feel it is there in the introduction, but casual readers may miss it.

The image quality and size of the olfactometer figure could be improved. Make sure the resolution and text are large enough to be clear in the published manuscript.

I feel this is important and difficult work that will help build the foundation to study this species behavior.

---

## [Author Response · AUTHORS RESPONSE TO REVIEWERS]

## Dear Dr Martins Jr.

I’m attaching the revised version of our article entitled “Behavioral responses of Anopheles darlingi (Diptera: Culicidae) to host odors: insights from vertical and horizontal olfactometers)” We have now completed the revision in the manuscript and responded (italics) to each of the referee’s questions or comments:

Editor: However, before the manuscript can be considered for publication, several substantial issues need to be addressed. These include concerns regarding the interpretation of behavioral differences between wild and F1 mosquitoes—particularly in light of potential differences in fitness and reproductive status—as well as the absence of key methodological details and incomplete figure presentation. We invite you to revise your manuscript thoroughly, addressing the methodological concerns, clarifying experimental conditions, and improving figure quality and consistency. Please also consider reinforcing the introduction with additional context on the biological relevance and experimental challenges associated with An. darlingi.

Authors: We agree with the editor’s comments and have restructured the text by incorporating new discussions. We added information in the introduction about the biological relevance of the species and the reproductive challenges of working with it. Regarding the behavioral differences between wild and F1 females, we have addressed them accordingly.

Referee 1: The authors report significant differences in odor attraction between the two mosquito groups across both olfactometer types. One striking pattern in the data is the low activity and host-seeking behavior in the F1 mosquitoes. This suggests that these first-generation lab-reared mosquitoes may suffer from reduced fitness, potentially as a result of stress from colonization, mating limitations, or rearing conditions. Their poor responsiveness calls into question whether they are valid representatives of An. darlingi behavior. A basic analysis of locomotor activity could help determine whether the F1 mosquitoes are capable of normal flight and orientation. Alternatively, establishing a more stable, multi-generational laboratory colony might yield more behaviorally robust individuals for comparison. As it stands, the comparison between wild mosquitoes and these F1 individuals—who may be behaviorally compromised—limits the strength and relevance of the study’s conclusions.

Authors: We understand the referee’s concern regarding the factors that may have led to the differences we observed between wild and F1 females. The difference in host-seeking behavior between the two lineages was expected, since the former are found mostly inseminated, as suggested by the literature and by dissections of a subsample of wild females in our study, while the latter are virgins. Considering our rearing conditions, similar to those used even in colonies that are now established, we believe the differences did not result from reduced fitness but rather from the physiological status of the lineages. We have included a paragraph in the discussion to address the impact of physiological state on host-seeking behavior. A locomotor activity study conducted with colonized An. darlingi showed that both inseminated and virgin females were highly active during the scotophase (Bastos et al., 2023). We discussed the differences between wild and F1 females using this information. In conclusion, with the changes made and considering the limitations of the study, we believe our most important conclusion—that olfactometers can be used for the study of An. darlingi—remains valid, and that, by considering and discussing the differences between field and F1 females, these results provide insights that can be applied in future assays with now-colonized mosquitoes. 

Referee 1: Another major point concerns the mating status of the mosquitoes. While the authors address this in the discussion, its implications for the results are severely underestimated. The authors state that “F1 females do not copulate spontaneously in the laboratory and require external stimuli like light or temperature changes.” In addition to the potentially lower fitness of the F1 mosquitoes, differences in reproductive state between wild and F1 females could explain the behavioral results and render direct comparisons between the two groups invalid. Dissection of F1 spermathecae (as done for wild mosquitoes) and oviposition assays could provide insight into how reproductively active F1 females were and thus clarify the impact of reproductive state on host-seeking behavior.

Authors: To provide context for the time when this work was conceived (2015), there were no established An. darlingi colonies available in Brazil (Araujo et al., 2019). Therefore, to obtain mosquitoes for medical entomology research, we routinely generated the F1 generation from wild-caught females collected in rural areas around Porto Velho, which were maintained in the insectary of the Fiocruz Rondônia Entomology Laboratory. Regarding the examination of spermathecae of virgin females, we considered it unnecessary since these were reared insects and not from a colony. This assumption was supported by the inability of An. darlingi to copulate in small cages (eurigamic behavior) unless colonies are established, which was not the case in our study. 

Referee 1: Wild mosquitoes were collected using human landing catches. This method biases capture toward females that are actively host-seeking, as clearly reflected in the olfactometer assays. However, this collection bias likely increases the behavioral differences observed between wild and F1 mosquitoes, which may be in very different physiological states.

Authors: Unfortunately, at the time this study was conducted, An. darlingi colonies were not yet available in Brazil, and the standard collection method for this species in the field—still in use today—is protected human landing catches. Indeed, as already mentioned and discussed in the paper, over 95% of field-collected females were inseminated, whereas F1 females in the laboratory were virgins, given that copulation does not occur under the rearing conditions used. Therefore, the differences between wild and F1 females likely resulted from their distinct physiological states and their specific responses to host odors, as we verified in our assays. Nevertheless, a recent study by Bastos et al. (2023), conducted with colonized An. darlingi, showed that both inseminated and virgin females were highly active during the scotophase. We discussed the differences between wild and F1 females using this information. 

Referee 1: While the authors provide detailed information on the rearing conditions for lab-reared mosquitoes, there is little description of the handling and maintenance of wild mosquitoes. How long after collection were they tested? Were females housed with males to increase the likelihood of mating?

Authors: They were tested approximately 48 hours after collection. Only females were collected in the field, since protected human landing catches do not allow male capture. As we reported in one sample, about 95% of field-collected females were already inseminated. Information about the handling and maintenance of wild mosquitoes has now been added to the Methods section.

Referee 1: A key piece of information missing from the manuscript is the dimensions of the olfactometers. The authors refer to the vertical olfactometer as a short-range host-seeking assay, but without dimensions, it is difficult to determine how short-range the assay truly is.

Authors: We apologize for the lack of detail. The dimensions have been included in the figures, based on the original articles used to construct the olfactometers. 

Referee 1: The legend for Figure 1 is incomplete. It refers to a part A and part B that are not labeled on the figure. While the panels are visually labeled, the legend does not explain what each part represents. This is especially problematic for the right panel. It appears that purified environmental air is divided among three olfactometers, but this is not described in the text or legend and should not be left to the reader’s interpretation.

Authors: The legend for Figure 1 has been revised, and we included the description of each part as requested. We removed the panel showing the three olfactometers to focus on the part that better explains the horizontal olfactometer itself.

Referee 1: Why was the control for the horizontal olfactometer an empty tube, rather than an unworn sock as in the vertical olfactometer? Based on the data presented, the horizontal olfactometer appears more effective in attracting F1 mosquitoes. However, the use of an empty tube as a control (instead of an unworn sock) may have increased the preference index observed in these experiments.

Authors: We followed the methodology proposed by Paixão et al. (2015), who used an empty tube as the negative control. In the vertical olfactometer assays, although the negative control is also an empty arena, we used a clean sock to ensure there was no interference from its components, but we observed no difference compared to the empty arena. Regarding the higher attraction of F1 mosquitoes in the horizontal olfactometer, we believe the empty tube was not responsible for the variation, since greater activation is expected in this device due to the airflow generating an odor plume, while in the vertical olfactometer odor diffuses passively to the upper tube. The horizontal olfactometer better simulates the natural behavior of mosquitoes, which involves oriented flight against the airflow. Despite these “differences,” the overall pattern—i.e., higher responses of wild mosquitoes compared to F1—was maintained. We added a paragraph at the end of the discussion to better address this issue. 

Referee 1: Panel A of Figure 2 appears to be missing. The plot shown lacks a label on the y-axis, and the meaning of the error bars is not stated.

Authors: We apologize. We submitted the wrong figure during submission. The correct figure has been added with the appropriate y-axis label.

Referee 1: In Figure 4, several statistical comparisons mentioned in the text are not reflected in the plot. Direct comparisons (e.g., Hands: wild vs. F1; Socks: wild vs. F1) should be clearly indicated with appropriate statistical annotations.

Authors: We found differences only in the overall comparison between wild and F1 mosquitoes, which was indicated by different letters next to the figure legend. We did not observe specific differences between the pairs Hands: wild vs. F1; Socks: wild vs. F1, and therefore no statistical notation was added for them. We revised the text to better reflect this result.

Referee 1: In line 22 of the abstract, the authors state “particularly in the context of Plasmodium-infected hosts.” It is unclear why this is mentioned, and it misleads the reader into expecting that host-seeking in infected hosts will be examined.

Authors: We removed this part of the sentence to avoid confusion for the reader.

Referee 1: Line 283: Of the mosquitoes tested, what proportion of females were positive for sperm in the spermathecae? This would provide valuable information on the reproductive state of the wild mosquitoes tested in the assays.

Authors: We did not analyze the spermatheca content of all wild females used in olfactometer assays. However, we were able to carry out this analysis in a subsample of wild females collected from the same areas and observed spermatozoa in 93/97 spermathecae. This information has been inserted in the last paragraph of the Results section, along with contextualization in the Discussion.

Referee 1: Is there a reason why the acclimation period was 5 minutes in the vertical olfactometer but only 1 minute in the horizontal olfactometer?

Authors: Few studies have been published testing the attractiveness of culicids in vertical olfactometers using human body parts. We based our method on the study published by Feinsod and Spielman (1979) on the attraction of Aedes aegypti to the human hand. That study used 5 minutes as a pre-test acclimatization period, and we adapted the exposure time from 7 to 10 minutes because personal observations made in the insectary suggested that An. darlingi is a more timid mosquito compared to Ae. aegypti. In the manuscript, there was a typographical error regarding the acclimatization period for the dual-choice olfactometer: where it read “one-minute acclimatization period,” it should have stated “ten-minute acclimatization period.” We followed the methodology of Paixão et al. (2015), with adaptations for An. darlingi. We corrected the acclimatization period in the Materials and Methods section.

Referee 1: Line 269: The authors state, “Overall, An. darlingi females were less attracted to olfactory stimuli compared to Aedes aegypti.” This comparison is not valid and should be removed, as it is based on data from different species and different studies, under potentially different experimental conditions.

Authors: We removed this comparison.

Referee 2: All generic names should be italicized. For example, line 58 - Plasmodium.

Authors: We carefully reviewed and corrected all scientific terms.

Referee 3: Please add a paragraph to the introduction explaining a bit more why Anopheles darlingi is difficult to work with. Some of this is in the discussion, but a brief summary in the introduction would give the reader more context. For example, the mating issues in the lab need to clearly stated in the introduction.

Authors: We added information about the biology of An. darlingi in the Introduction.

Referee 3: I would also add a sentence or two more on the importance of the species. I feel it is there in the introduction, but casual readers may miss it.

Authors: We agree with Referee 3 and added information in the introduction on the importance of the species in Brazil and its activity patterns.

Referee 3: The image quality and size of the olfactometer figure could be improved. Make sure the resolution and text are large enough to be clear in the published manuscript.

Authors: We improved the quality, size, and information of the olfactometer figures.

Thank you for your time, help and interest. Many thanks to referee 1 for his/her detailed revision and suggestion to improve our manuscript. This is an important publication for us; therefore, I look forward to hearing from you.

Sincerely,

Alexandre Silva

---

## [Reviewer Report · REVIEWERS COMMENTS]

## REVIEWER #1

The authors addressed all my concerns and I recommend this manuscript for publication. Please ensure that Anopheles darlingi is italicized in all figure legends.

## REVIEWER #2

I feel the authors have addressed my questions adequately. I have two further suggestions to improve the manuscript.

1. The authors should add the sentence starting on line 51 to the previous paragraph. It is better not to have a paragraph fragment.

2. In figure 4, the statistical significance (letters) should be put next to the data not in the legend. The letters should also be capitalized.

Otherwise, the manuscript is ready for publication.

## REVIEWER #3

No comments.